# Moore's Law revisited through Intel chip density

**David Burg** [1,2,3☯] *, **Jesse H. Ausubel** [3☯]

**1** Eastern Research and Development Center, Ariel, Israel, **2** Tel Hai Academic College, Upper Galilee, Israel, **3** Program for the Human Environment, The Rockefeller University, New York, NY, United States of America

☯ These authors contributed equally to this work.
* davidbu@ariel.ac.il

**Data Availability Statement:** All relevant data are within the manuscript and its Supporting Information files.

**Funding:** The authors received no specific funding for this work.

## Abstract

Gordon Moore famously observed that the number of transistors in state-of-the-art integrated circuits (units per chip) increases exponentially, doubling every 12–24 months. Analysts have debated whether simple exponential growth describes the dynamics of computer processor evolution. We note that the increase encompasses two related phenomena, integration of larger numbers of transistors and transistor miniaturization. Growth in the number of transistors per unit area, or chip density, allows examination of the evolution with a single measure. Density of Intel processors between 1959 and 2013 are consistent with a biphasic sigmoidal curve with characteristic times of 9.5 years. During each stage, transistor density increased at least tenfold within approximately six years, followed by at least three years with negligible growth rates. The six waves of transistor density increase account for and give insight into the underlying processes driving advances in processor manufacturing and point to future limits that might be overcome.

## Introduction

It has been observed that the number of semiconductor components on a silicon chip increases exponentially and is expected to stop growing only when uncertain limits have been reached [1]. The observed trend slowed from a doubling in the number of components per chip every year to doubling every two years, with an intermediate doubling time of 18 months [2]. These observations are generally referred to as "Moore's Law" [3], a benchmark that has become a largely undisputed, though perhaps misunderstood, rule for the microprocessor industry [4, 5].

Sigmoidal models have been shown to be compatible with technological evolution, even in the context of Moore's Law of transistor performance [6–8], giving rise to decreasing growth rates as a technology matures. S-curves can describe the growth of technological performance [9–11]. Further, these patterns have been recognized in innovation generally, including technology life-cycles and learning systems [12–14]. The ability of the simple logistic model to describe this process may be due to intrinsic technological and physical factors, as well as economic forces constraining unfettered increase in complexity. Technological progress has also

**Competing interests:** The authors have declared that no competing interests exist.

been identified with a quasi-fractal wavelet process conceptualizing growth as an agglomeration of distinct subprocesses [15].

Indeed, many systems of increasing complexity and information exhibit discontinuous multiphasic trends [16]. The hypothesis of linked S-curves has been put forth by Foster [17] and Christensen [18]. It is clear that technological evolution, hypothesized here to include transistor miniaturization, is discontinuous and that new designs and processes are distributed unevenly through time in "innovation waves" [19]. Technological evolution frequently displays these more complex kinetics with a tendency to saturate because of constraining factors [20–23], and this pattern is also reported for semiconductor performance [24, 25]. Logistic component analysis [26], as well as rate analysis, of empirical data may be able to discriminate the timing of important technical improvements directly from the data, elucidating the main trends in the miniaturization of transistors. These descriptive models do not attempt to explain the underlying mechanisms of increasing technological performance *per se*, though they can give insight into evolution of the complex system [27].

Most research has accepted Moore's assumption of the exponential doubling of computer processor complexity, defined as the number of units per chip. However, increases in transistor count may be coupled with increases in chip size (die area); that is, more transistors can be added to a processor by increasing its size (area or volume). Recently, a new definition of transistor density has been suggested [28], and we propose to reexamine processor evolution by examining growth in the number of transistors per unit area, accounting for changes in chip size. Analysis focusing on density of state-of-the-art products allows development of an envelope function as an indicator of the fabrication capabilities and techniques. This paper tests whether simple exponential growth is consistent with the historical time-series data or if a more complex model would provide a better description of this technological, as well as economic, phenomenon. Information-based methodologies of parameter optimization and model selection are adopted for optimal statistical efficiency.

## Materials and methods

### Data compilation

Data for early integrated circuits were collected from Fairchild Semiconductor International [29–36] and for modern central processing units (CPUs) from Intel (http://ark.intel.com) between 1959 and 2013, for consistency. For each product, information was recorded for CPU type (desktop/mobile), release date, clock frequency (MHz), fabrication process (nm), number of transistors, and circuit area ($mm^2$). Technical information on newer CPUs has not been reported by Intel since 2014, though information for high-end products is only available from various online but unofficial publications [37–41].

We focus on Fairchild and Intel data because they form the longest publicly available time series, and because of Gordon Moore's experience first as director of research and development at Fairchild and later as an executive of Intel where he proposed his Law in 1965. A complementary analysis could span chips from Texas Instruments, but Moore's counterpart at TI, Patrick Hagerty, earned fame for a 1964 prediction of production of logic gates a decade hence that turned out to be a large underestimation [42]. Once a major manufacturer of chips, IBM has for some years outsourced its high-volume chip production to Samsung, which started production in the 1980's, as did competitors Hynix (1983) and TSMC (1987). Compiling a comprehensive database including other American as well as European and Asian producers would be a considerable task.

To represent the trend in the state-of-the-art technology, the highest density product per year will illustrate the industry's capability to increase transistor miniaturization, deriving the

time-domain envelope of fabrication capabilities and density of CPU transistor technology. This methodology was chosen because data are available only for top-end models during the early years of IC manufacturing, even though less sophisticated products undoubtedly existed. More important, including products based on older manufacturing technologies during the same timeframe would compromise the ability to determine the constraints of the system. Finally, this will provide an envelope function for transistor miniaturization dynamics. See S1 Table for the data conforming to the inclusion criteria in this investigation.

Industry experts have traditionally tracked change in the number of transistors per chip as the variable of integrated circuit evolution. Yet, this omits changes in chip size and accordingly does not implicitly elucidate transistor miniaturization trends. Like Ferain et al. [43], we evaluate processor evolution here by transistor density, defined as the average number of transistors per unit area, and thus spotlight miniaturization.

## Mathematical models

Moore's Law states that the number of transistors increases inexorably and that growth rates may change at different times; this is mathematically congruent with a stepwise exponential function:

$$T(t) = a_i e^{r_i t} \begin{cases} r_1, a_1 & \tau_1 < t \leq \tau_2 \\ r_2, a_2 & \tau_2 \leq t < \tau_3 \end{cases} \tag{1}$$

where $T$ is the transistor density variable, and the growth phase is denoted by $i$. The growth rate constant ($r_i$) encapsulates research and development resources, fabrication techniques, and other factors at different phases during transistor evolution ($\tau_i$). A semilog transformation will linearize Eq (1), and computation of doubling times is straightforward ($t_2 = \ln[2]/r_i$). Modeling processor evolution, with more complex kinetics and a tendency to saturate because of constraining factors, can be achieved with the generalized multilogistic model [44] to describe consecutive waves of technological development ($n$):

$$T(t) = \sum_{i=1}^{n} \frac{K_i}{1 + e^{-r_i(t - \tau_i)}} \tag{2}$$

which has two shape parameters: the intrinsic growth rate constant ($r_i$) and the saturation level ($K_i$) for each growth phase ($i$). The "characteristic time" ($\Delta t_i = \ln[81]/r_i$) represents the time for the system to grow from 10% to 90% saturation [45]. The midpoint ($\tau_i$) is a location parameter determining the time of the inflection point when the trend has reached half-saturation. It partitions the curve between concave and convex growth patterns, and the model converges asymptotically to the saturation value (Fig 1A). The advantage of this model is that it encapsulates a system's tendency to evolve rapidly, followed by maturation and saturation. In this context, the industry's ability to miniaturize transistors can be modeled, where $r$ and $K$ may vary over time [46]. After $\log_{10}$-transformation, the logistic function no longer appears sigmoidal. Only the left tail is linearized since the exponential part dominates the behavior of the logistic at low densities (Fig 1B), unlike the exponential model, which is linear over the entire range. The individual sigmoidal growth pulses may be decomposed into their constituent curves, which are then linearized:

$$-\ln\left(\frac{F}{1 - F}\right) = r_i(t - \tau_i) \tag{3}$$

where $F = T/K_i$ for each growth phase $i$ (assuming $T < K_i$) and the new scale can be approximated to the percent of the growth curve [47].

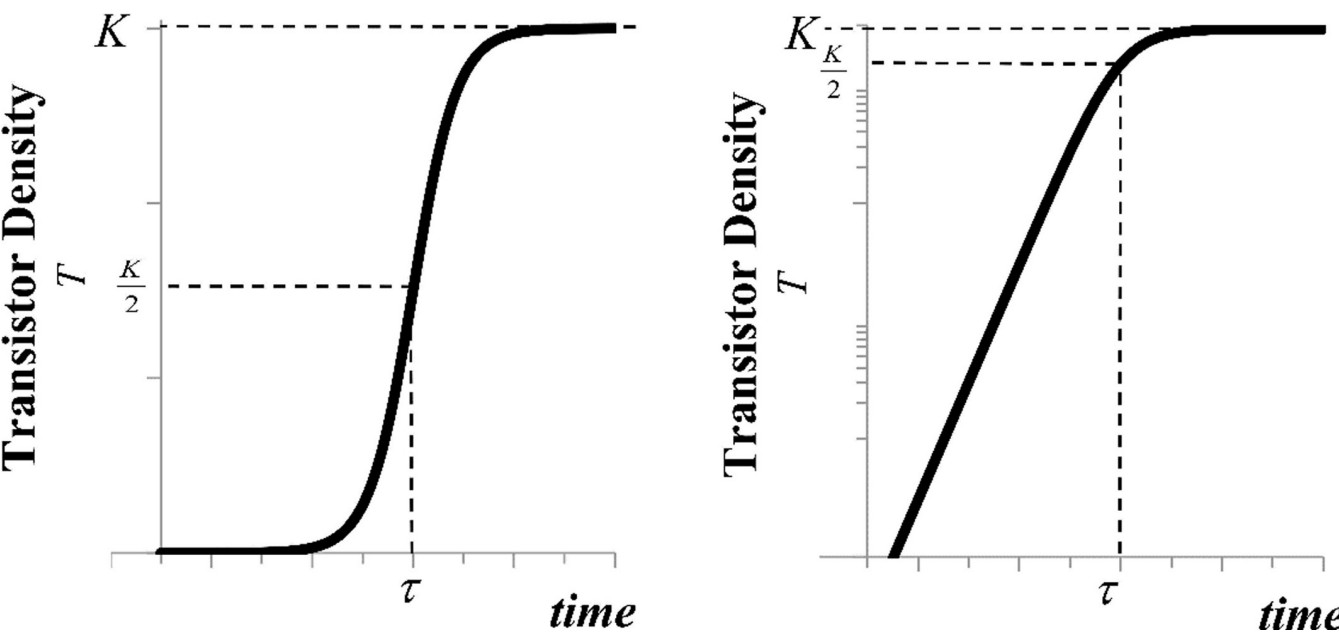

**Fig 1. The general shape of the time-dependent logistic model compared with the curve following semilog transformation.** A) The logistic model on a linear scale. B) The semilog transformation of the logistic model. Note the nearly exponential behavior of the initial phase of the function. The midpoint half-saturation point ($\tau$) and asymptotic saturation ($K$) are shown.

### Parameter estimation, statistical analysis and model selection

Initial parameter values for initiation of the nonlinear fitting algorithm can be obtained directly from the estimated data. A pattern-recognition algorithm was developed to detect significant increases in the slope between three data points ($r > 0.35$ yr$^{-1}$) followed by a reduction of the slope such that the slope approaches horizontal ($t_2 \leq 0.5$ yr) over a span of at least four consecutive years, giving preference to steady states. Saturation points ($K$) were determined at the end of each interval. Since the logistic model converges to the exponential model at $T \ll K$, maximal estimates for $r$ were calculated from slopes of log$_n$-transformed data preceding inflection points. To delineate the changes in growth rates, the acceleration properties of the data can be approximated using the finite differences method ($d^2T/dt^2$). An inflection point, where acceleration becomes negative, will be observed when the second derivative crosses from positive to negative. However, this analysis inherently increases noise in the result, and smoothing was performed to increase the signal-to-noise ratio [48]. To confirm the discrimination of multiple peaks in growth rates, the software package Automatic Maxima Detection [49] was used algorithmically to identify peaks in rate of change embedded in the data to confirm these results. These allow determination of minimal initial estimates for the numerical integration and fitting the data to the mathematical models.

Optimized parameter values were then obtained using a simulated annealing Monte Carlo–based genetic algorithm [50]. Briefly, a population of theoretical curves is constructed from randomly generated parameter sets, and each fit is given a fitness score based on the likelihood function:

$$L = \sum_{i=1}^{n} P(x_i - X|\theta) \tag{4}$$

where $L$ is the likelihood of observing residual ($x_i$—$X$) assuming a normal distribution ($\theta$),

where $x_i$ is the data point at time $i$ and $X$ is the expected value for $n$ data points. The parameter space of the subset of fits with the lowest values (best fits) is then used for multiple iterations. Log$_{10}$-transformed data stabilize the variance during the fit [51]. Confidence intervals for the parameter values were constructed using the bootstrap method [52]. Pearson correlations, tests for heteroscedasticity and autocorrelation (Breusch-Godfrey and Durbin-Watson tests, respectively) were performed on the linearized data [53]. Goodness-of-fit was evaluated from the root mean square (RMS) and mean absolute percentage error (MAPE). The R-square statistic was not calculated as it is inappropriate in nonlinear systems [54]. The corrected Akaike information criterion (AICc) was used for to appropriately perform model selection among competing models of differing complexity [55].

## Results

### Reproduction of Moore's Law

Initially, Moore's Law is reproduced by tracking the number of transistors per chip ($T$) as a function of time with two distinct phases. Doubling times of 14 and 25 months are similar to Moore's estimates from 1965 and 1975, respectively. Concomitantly, processor die area ($A$) also exhibits exponential slopes with a doubling time of 8.3 years (r = 0.94, P< 0.001). Regressing these two variables indicates that chip size is coupled with the number of transistors through a power-law relationship (Fig 2). This gives evidence that the "number of transistors per chip" may be a biased descriptor of circuit density. Increases in transistor counts do not explicitly indicate miniaturization of transistors, because increasing the number of transistors on a processor can also be achieved by increasing the die size.

### Transistor density

Based on this finding, we define here transistor density as the number of transistors per unit area. These rescaled data were then fit to the Moore's Law stepwise exponential model (Eq 1). The results are shown in Fig 3. The resulting doubling times are 17 and 33 months, for the first and second phases, respectively (Table 1). These values are nearly 30% less rapid than the 12 to 24 months reported by Moore (see Introduction). However, the data exhibit significant heteroscedasticity (P = 0.024) and autocorrelation (P<0.001). Indeed, this bias of the data to be above or below values predicted by the model, especially all data since 1999 being well above the expected value, implies that the stepwise exponential model is unsuitable to these data.

### Sigmoidal trends of processor evolution

The density of transistors was then fit to Eq (2), resulting in a well-defined bi-logistic trend (Fig 4A). Interestingly, both phases have characteristic times ($\Delta t_i$) of 9.5 years. Midpoints of these distinct growth curves occurred circa 1979 and 2008, with approximately 30 years separating them. The first growth pulse saturated at approximately half the saturation of the log-cumulative distribution (Table 2). Decomposition of the linearized bi-logistic into its component phases is shown in Fig 4B. Values for the model selection criteria are lower for the bi-logistic model (AICc = -40) than for the stepwise exponential model (AICc = -26).

According to Moore's Law, the data should exhibit the distinctive acceleration properties of an exponential growth curve. However, the data exhibit fluctuations with clear decelerations trending through multiple inflection points, where the second derivative crosses from positive to negative, reaching multiple minima (Fig 5). These, of course, are followed by rapid accelerations.

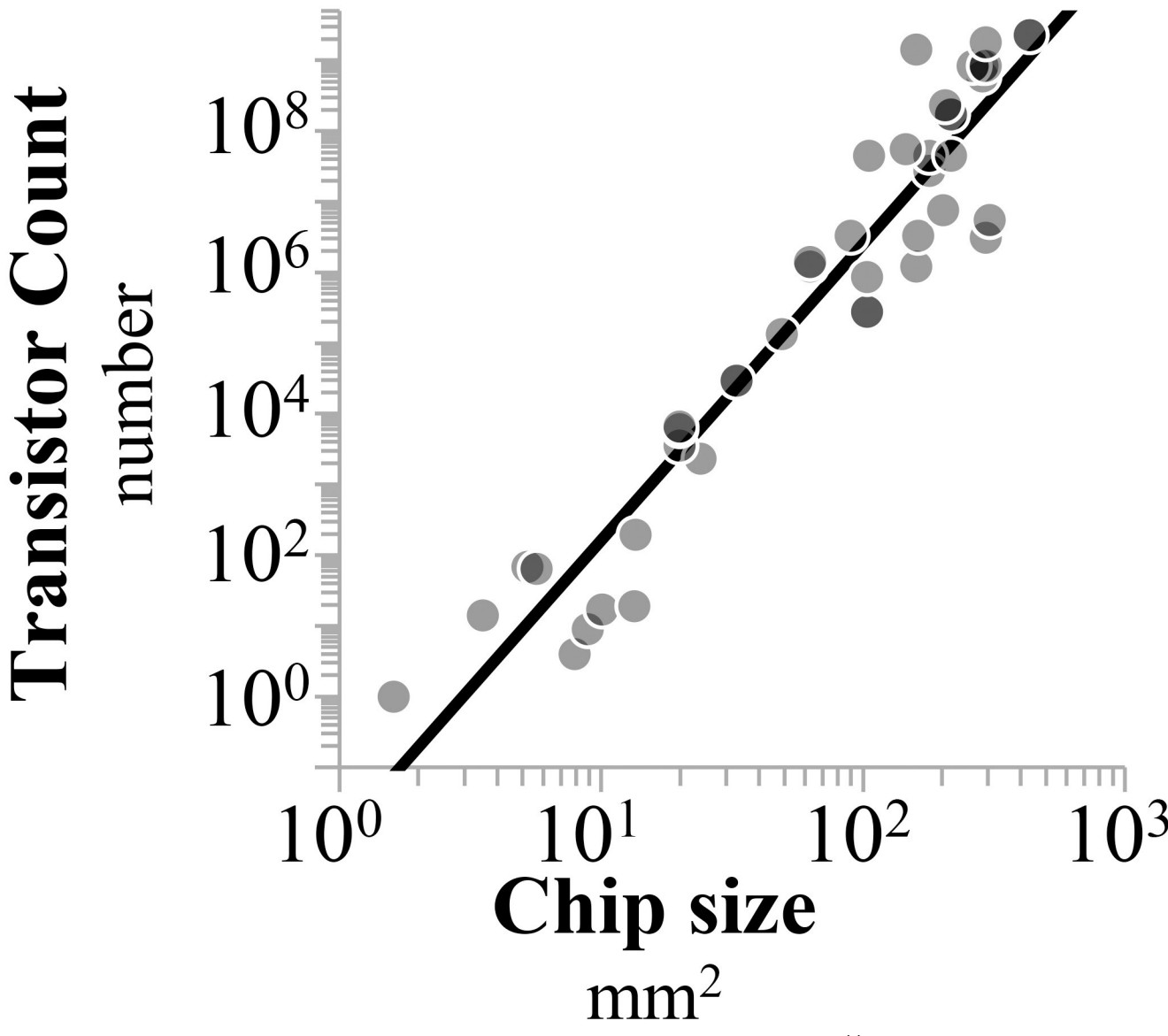

**Fig 2. A power-law relationship between state-of-the-art processor size and the number of transistors ($T \propto A^{4.4}$, P< 0.001).**

These multiphasic dynamics embedded in the CPU transistor density data indicate six significant periods of growth delineated by rapid increases of growth rates in transistor density followed by stable periods of at least three consecutive years. Fig 6 shows the decomposition of the individual logistic wavelets from information derived above. Table 3 summarizes the multilogistic model parameter values. Durations of these growth phases spanned 7 to 11 years (mean = 9 yrs). Mean growth rate constants of 0.8 $yr^{-1}$ correspond to a mean characteristic time of 6 years, with typical transition between phases of approximately tenfold in transistor density. Therefore, multiple logistic trends are indicated with sequential patterns of technology substitution.

## Discussion

The work presented here attempted to test the assumption of simple exponential trends in computer processor technology. Trend lines A-B and D-E in Fig 6 represent the sequential

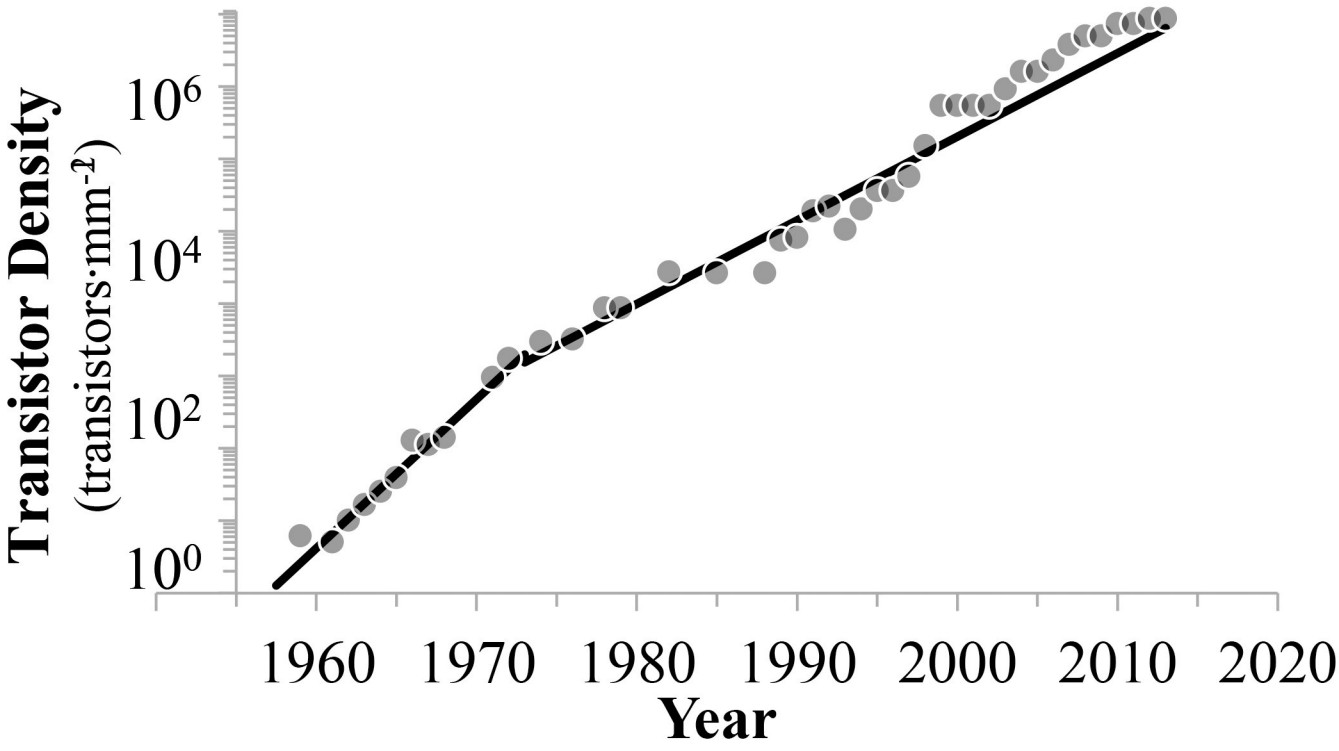

**Fig 3. State-of-the-art integrated circuit density (transistors·mm$^{-2}$) per year.** The two exponential growth phases are characterized by doubling times of 17 and 33 months, respectively. Data are heteroscedastic and autocorrelated, consistently underestimating all data since 1999.

patterns where technologies driving the first logistic curve saturate and are replaced by new ones [56]. Interestingly, two trends originate together but are divergent (Fig 6B, 6C, 6E and 6F), perhaps indicative of self-propagating growth in performance [57, 58]. During each stage, transistor density increased at least tenfold within approximately six years, followed by at least three years with negligible growth rates. Rapid transistor miniaturization is achieved during only two-thirds of the history of the transistor. This makes sense from an economic point of view with the need to increase revenue through continued production of products based on established technologies along with the introduction of newer products. This allows economic returns to be realized from the exponentially growing investments in research and development required by each new pulse of advances [59]. Waves of miniaturization (denser and even physically smaller chips) may have multiplied markets as much as the growing chip size measured in units, such as number of transistors.

The transient logistics of CPUs depicted here point to the technological advances driving the waves that make up the process. The first commercial planar transistor developed at

**Table 1. Stepwise exponential model parameter values.**

| Phase | Phase beginning | Phase duration | Doubling time |
|:-----:|:---------------:|:--------------:|:-------------:|
| $i$ | $\tau_i$ | years | $t_2$ |
| 1 | 1959 | 13 | 17 |
| 2 | 1973 | 42 | 33 |

RMS = 0.071, MAPE = 1.82, AICc = -26, P< 0.001.

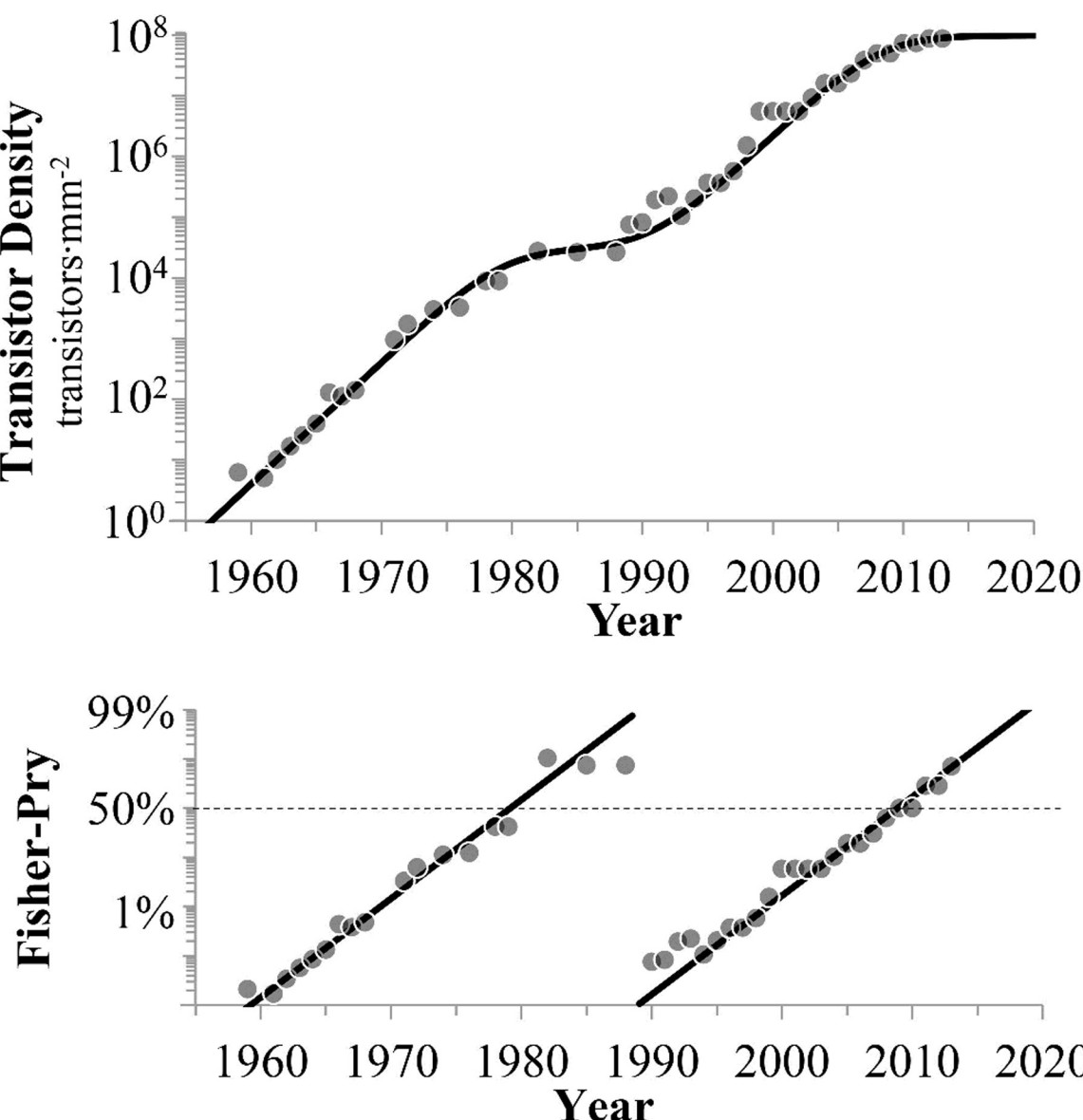

**Fig 4. The temporal trend of transistor density.** A) The data exhibit a bi-logistic curve. B) Decomposition and linearization of the individual trends are depicted as percent of growth for each of the two phases. Parameter values are shown in Table 2.

**Table 2. Bi-logistic parameter values.**

| Phase | Characteristic time | Midpoint | Saturation limit |
|---|---|---|---|
| $i$ | $\Delta t_i$ | $\tau_i$ | $K_i$ |
| | years | year | log transistors·mm$^{-2}$ |
| 1 | 9.5 [8.5–10.5] | 1979 [1978–1981] | 3.0 [3.45–3.50] |
| 2 | 9.5 [9.0–10.0] | 2008 [2007–2009] | 7.0 [6.96–7.03] |

RMS = 0.034, MAPE = 0.028, AICc = -40, P<0.001.

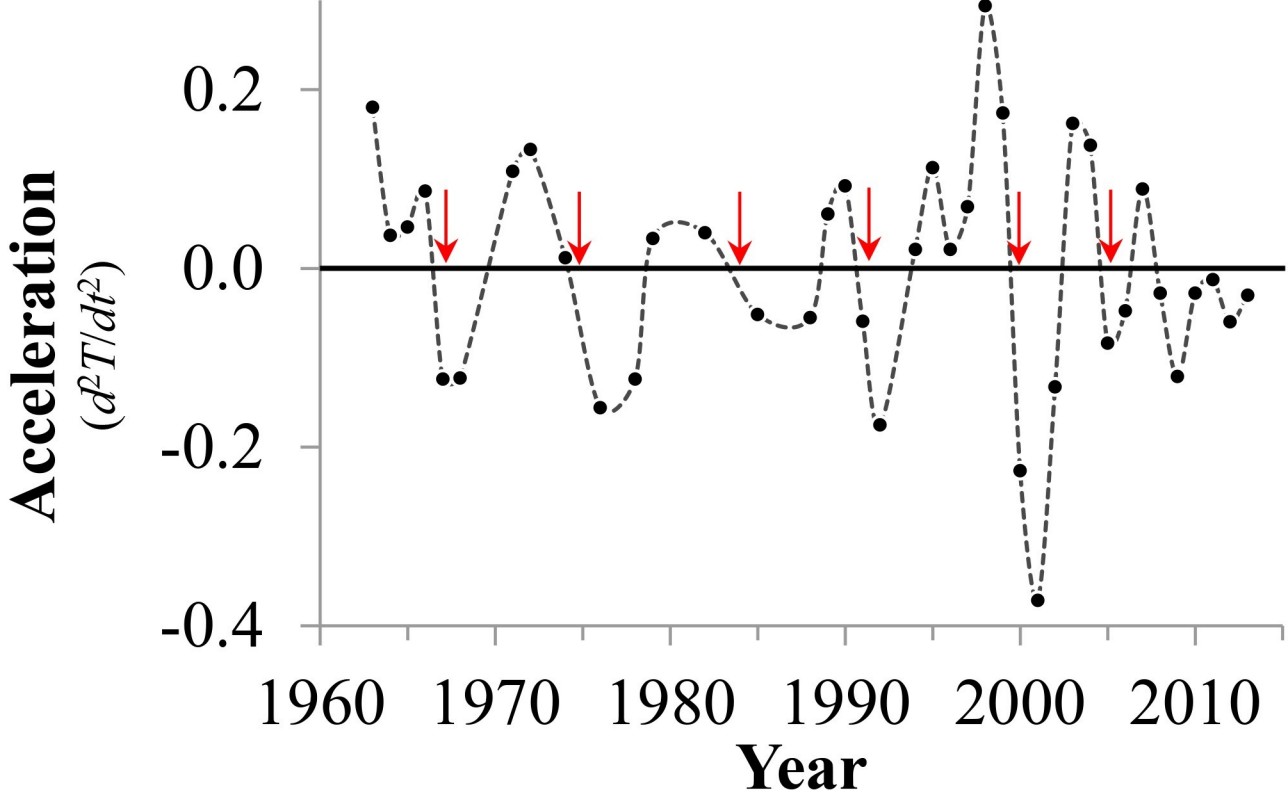

**Fig 5. Approximated second derivative exhibits complex acceleration and deceleration patterns in the data.** Arrows indicate inflection points where growth rates decline.

Fairchild Semiconductor in 1959 [60] was based on the demonstration of the silicon transistor and adaptation of photolithography techniques, both developed at Bell Labs in 1954 and 1955, respectively [61–63], as the basis for the first phase (line A). The metal-oxide-semiconductor field-effect transistor (MOSFET), the foundation for all future transistor technology, was patented and commercialized by General Microelectronics in 1964 [64], perhaps accounting for the initiation of the second logistic wavelet (line B). Silicon gate technology (SGT) was first implemented by Intel [65] and was the precursor for all subsequent microprocessors, beginning with the 4004 and 8080 released in 1971 [66] concomitant with the beginning of the third wave (line C). High-density, short-channel MOS (HMOS), patented in 1977, substantially increased transistor density for the 8086 released in 1978 [67]. The 80486, which debuted in 1989, allowing substantially more transistors, permitting the integration of complex circuitry, such as 8 kB cache and a floating-point math coprocessor (line E). Deep–UV excimer laser lithography, demonstrated in 1982 [68] was commercially deployed during the 1990s [69], perhaps indicating the sixth wavelet (line F), since all processors released since 1998 were manufactured based on this technology. The technologies underlying the third and sixth waves were perhaps the most important during transistor evolution, for the development of the industry for two decades each. While this gives a brief tour of some key causal developments, we refer readers interested in more details on this subject to books by Seitz and Einspruch [42] and Lojek [32] as well as the IEEE article 25 Microchips That Shook the World [70] and the website Computer History Museum on The Silicon Engine: A Timeline of Semiconductors in Computers [71].

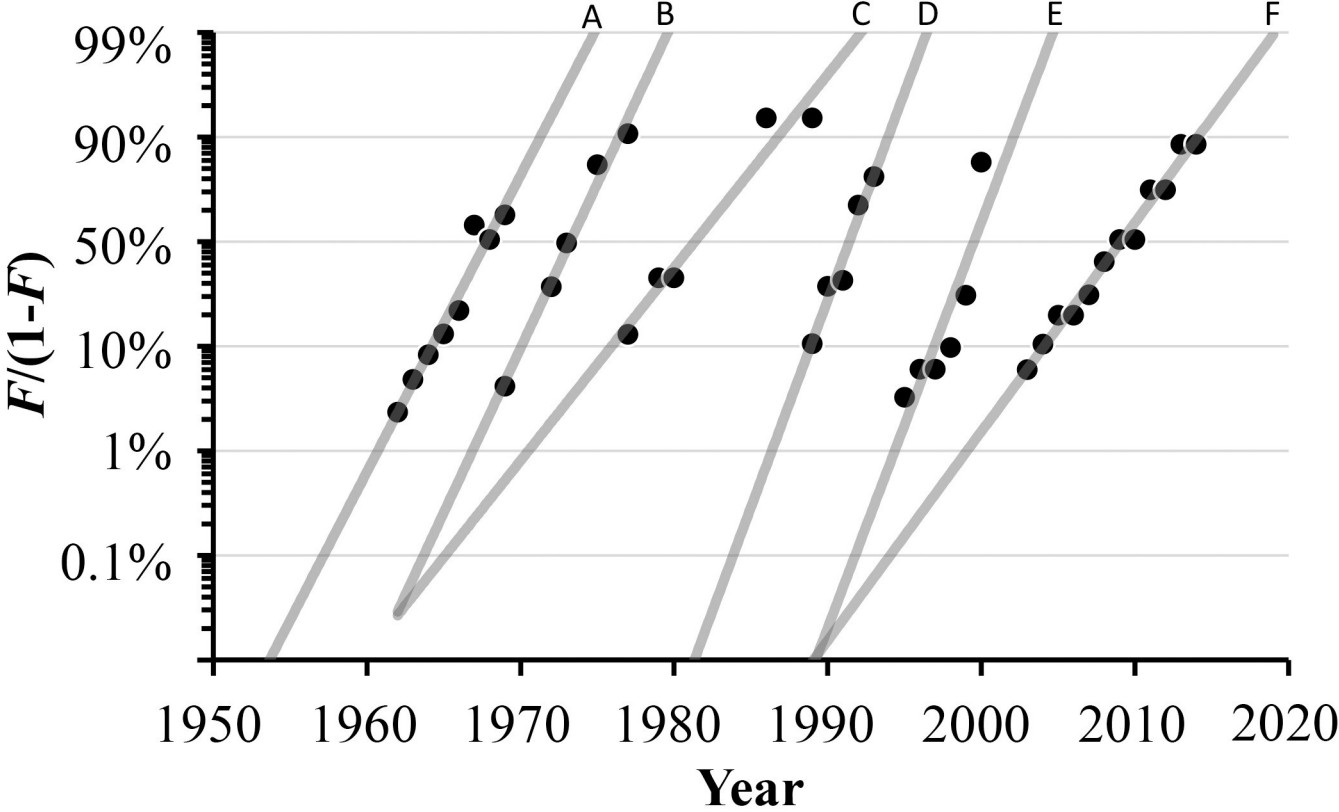

**Fig 6. Decomposition and linearization of data into corresponding individual loglet trends highlighting distinct phases of transistor evolution.** See Table 3 for fitted model parameter values.

An added advantage of the methodology presented here is the ability to track changes in mean transistor size, which is the reciprocal of the density function; this is dissimilar to the conventional technology node process as defined by the "minimum feature size" [72]. Data since 2000 exhibit a significant deceleration in miniaturization trends (Fig 7) while even relatively important advances have only kept transistor miniaturization on this decelerating trajectory, such as Intel's 3D tri-gate technology. Throughout the last two decades, advances in transistor miniaturization have slowed substantially and seem to signify a departure from the

**Table 3. Bi-logistic parameter values.**

| Phase | Characteristic time | Midpoint | Saturation limit |
|---|---|---|---|
| $i$ | $\Delta t_i$ | $\tau_i$ | $K_i$ |
| | years | year | log transistors·mm$^{-2}$ |
| A | 6.1 [6.0–6.2] | 1966 [1965–1967] | 1.26 [1.2–1.4] |
| B | 5.5 [5.4–5.6] | 1972 [1971–1973] | 2.54 [2.5–2.6] |
| C | 7.5 [7.4–7.6] | 1980 [1979–1981] | 3.45 [3.4–3.5] |
| D | 5.3 [5.2–5.4] | 1991 [1990–1992] | 4.49 [4.4–4.6] |
| E | 4.9 [4.8–5.0] | 1999 [1998–2000] | 5.83 [5.7–5.8] |
| F | 9.5 [9.4–9.6] | 2008 [2007–2009] | 9.95 [6.9–7.0] |

RMS = 0.008, MAPE = 0.019, AICc = -46, P<0.001.

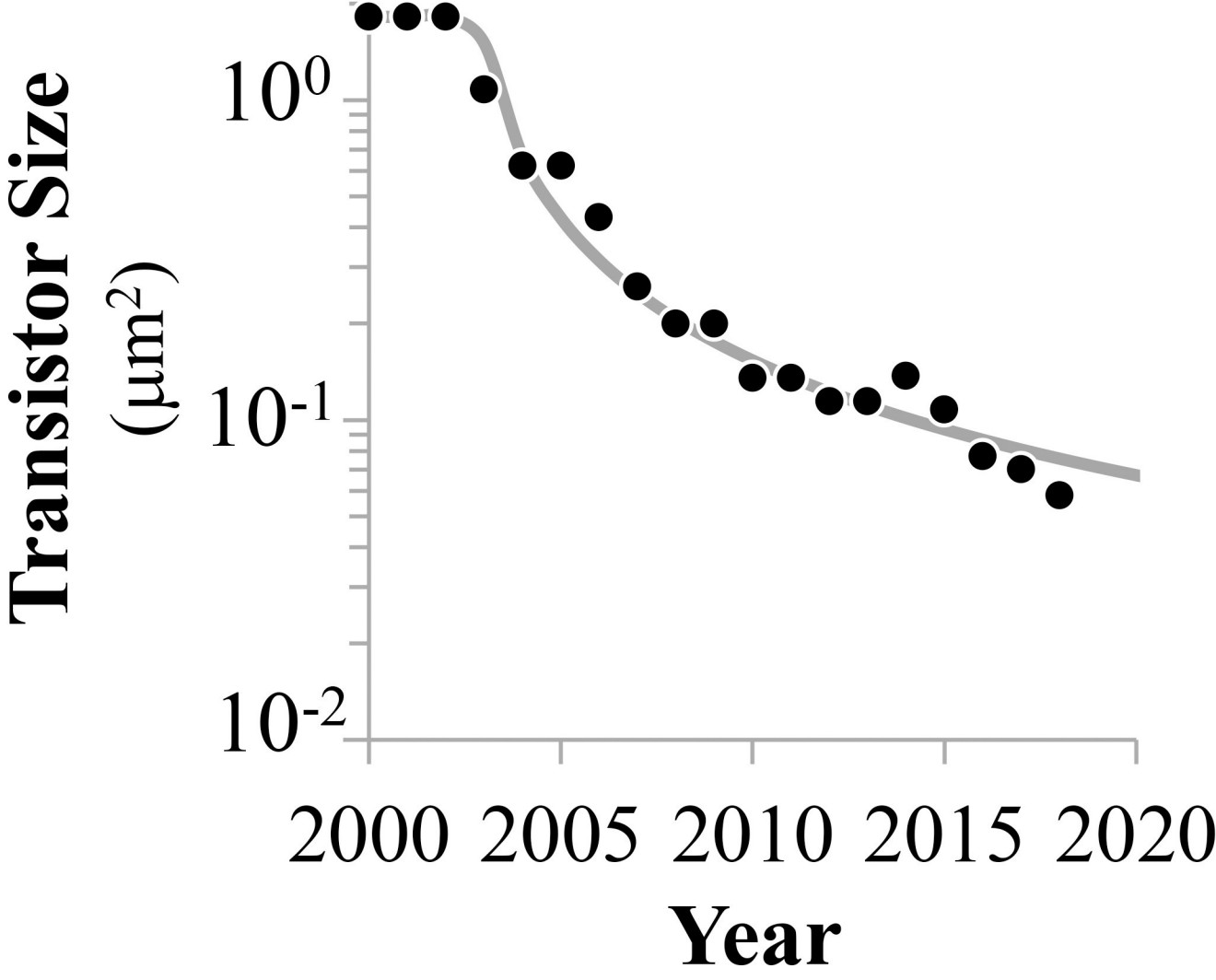

**Fig 7. Decreasing mean transistor size since 2000.**

International Technology Roadmap for Semiconductors, assuming of course the continuation of this trend. This may also explain current difficulties in attaining fabrication techniques for 10nm, 7nm, or smaller processes. Manufacturing challenges have increased because of the "subwavelength gap" at each technology node [73]. Indeed, the last decade has shown that strained SiGe [74], high-k metal-gate transistors [75], Resolution Enhancement Technologies [76], and FinFET circuitry [43] have allowed continued increases in transistor density, although at a markedly slower linear scaling rate, as indicated.

The fundamental limits imposed on integrated circuits by known laws of physics have been calculated and estimated to be many orders of magnitude beyond current fabrication capabilities [77]. It has been suggested that data parallelism with increasing core counts per chip are able to double computing performance [78]. This seems unlikely as only certain tasks can be multithreaded with efficiency and then the improvement is not in speed but in concurrency of instructions. Other physical constraints on the system, such as thermal limits, are nontrivial [79]. These, as well as the economic limitations imposed by exponentially growing considerations may be at least as important as the physical limits [80, 81]. Notwithstanding significant

technical and scientific advances, the current phase in transistor miniaturization is experiencing decay in miniaturization rates, characteristic of the dynamics around the midpoint of the logistic model.

Intel ceased reporting transistor counts and die sizes for their products in 2014. For the past five years, only estimates are available for the information needed for this analysis, and these cannot be confirmed by the authors. Nonetheless, the estimates seem to validate the model and provide evidence for the performance of this methodology with no significant change in processor miniaturization trends. Even as an increase in the number of transistors is reported, the size of the processor die also continues to increase, apparently offsetting the slowing improvement in transistor density.

Based on the analysis demonstrated here, the next growth impulse in transistor miniaturization is due. What challenges would elicit a market large enough to justify the effort to overcome the difficult obstacles ahead? A strong possibility is the desire for artificial intelligence (AI) to emulate biological intelligence, including capacity to acquire new knowledge from a sequence of experiences to solve progressively more tasks, and to offer empathy and imagination. AI researchers have advanced algorithmically and increasingly demand hardware to process quantities of data and train AI models. Designers embed significant amounts of fast memory in larger and larger chips to handle AI training algorithms requiring huge amounts of communication but relatively easy computation. For example, Xilinx announced (for the moment) the world's largest field-programmable gate arrays with 9 million system logic cells and 35 billion transistors, the highest logic density on a single device yet built, to enable development of complex algorithms for machine learning, video processing, and sensor fusion [82].

The startup company Cerebras has touted the largest chip ever built, the Wafer-Scale Engine, 56 times the size of the largest graphical processing unit (GPU), which has dominated computing platforms for AI and machine learning. The wafer-scale chip has 1.2 trillion transistors, embeds 400,000 AI-optimized cores (78 times more than the largest GPU), and has 3,000 times more in-chip memory [83].

"5G" infrastructure promises multi-Gbps peak data speeds, ultra-low latency, more reliability, massive network capacity, increased availability, and a more uniform user experience to more users. The required infrastructure, including access points such as mobile devices, cars, drones, and the Internet of Things will sum to an enormous amount of hardware. For example, at least 10 cameras and 32 sensors per car will be needed for high levels of autonomous driving. Given expected growth in data, 5G networks will reach capacity around the end of this Zettabyte decade, and around 2030 we enter the Yottabyte era for total global data generation. In short, the world of AI demands tremendous amounts of processing power to solve complex problems, even if using relatively simple algorithms, and silicon optimized for machine learning.

Some researchers have predicted that the silicon-based Moore's Law will fail in approximately 2020 [84]. Our analysis suggests that this possibility may have been underestimated. Transistor density evolution of the past decade conforms to a linear trend connoting slow and incremental advances, but also signifying a substantial departure from Moore's exponential law. Here we take a speculative leap by making a hyperlogistic from the inflection points of the six identified logistic wavelets (Fig 8). In effect, we have concentrated each of the six growth pulses in Fig 6 into a single point. The result is a projection that transistor density evolution may indeed saturate, but after one or possibly two more pulses.

The observation of a deceleration and plateau in transistor miniaturization suggests inherent technological difficulties, as well as possible strategies. Increasingly, the number of transistors is not necessarily optimal where constraints in energy efficiency or cooling require that

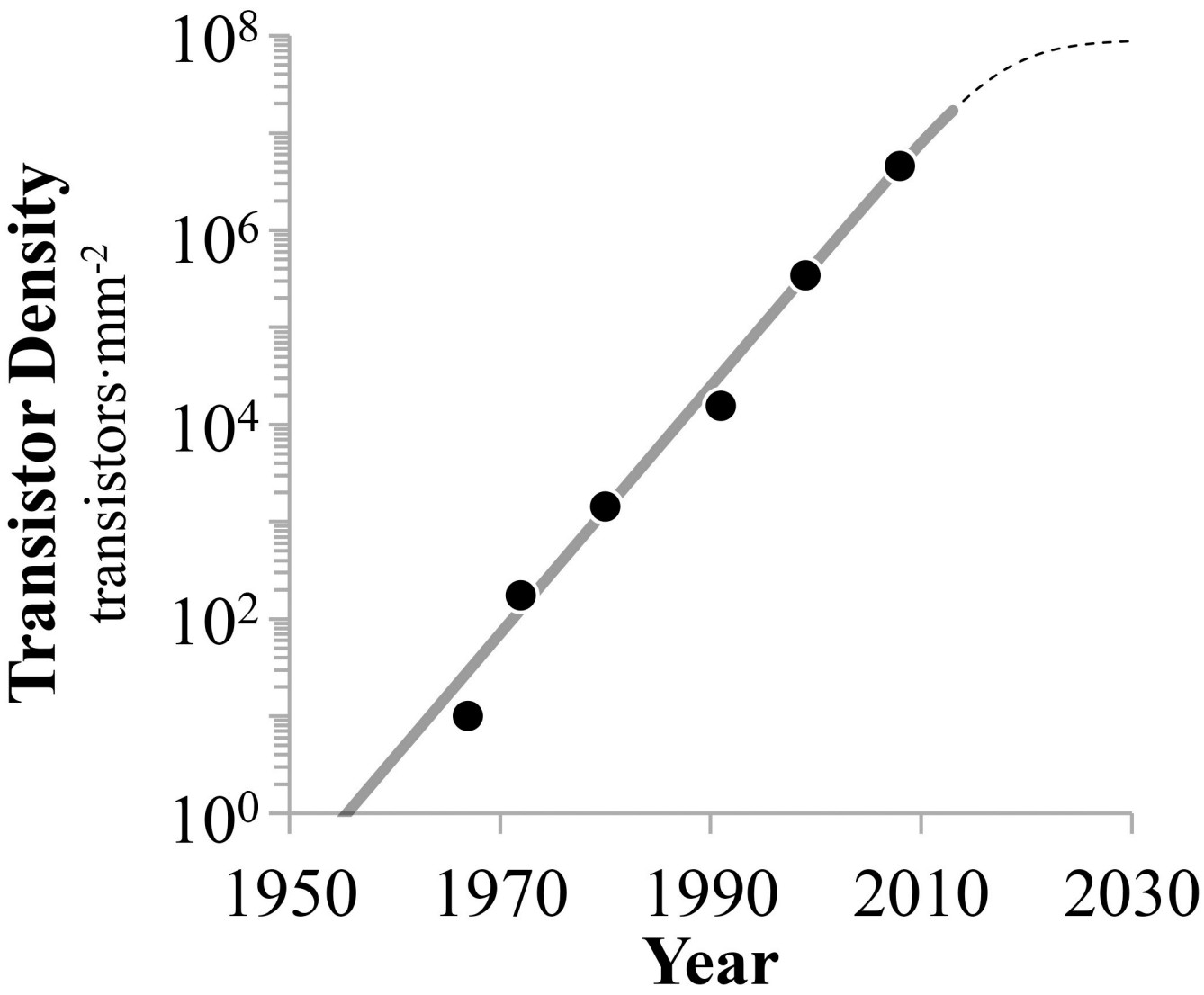

**Fig 8. A "hyperlogistic" function fitted to the inflection points of the six identified logistic wavelets.**

parts of an integrated circuit be powered off during operation, the design challenge referred to as "dark silicon" [85]. How can the computer industry continue to grow and innovate? As suggested above, advances in software optimized for parallel computing allowed by multicore processors are an important avenue of maximizing processing power [86]. Work has been under way for over a decade to realize immersive excimer laser and EUV metrology [87]. New research is under way into nanotransistors [88, 89] and single-atom-transistors [90], while another possibility may be quantum computing [91]. In 2019 Alphabet claimed a breakthrough in quantum computing with a programmable supercomputing processor named "Sycamore" using programmable superconducting qubits to create quantum states on 52 qubits, corresponding to a computational state-space of dimension $2^{53}$ (approximately $10^{16}$) [85]. The published benchmarking example reported that in about 200 seconds Sycamore completed a task that would take a current state-of-the-art supercomputer about 10,000 years. Each of these technologies has advantages and hurdles that must be overcome to realize a new rapid growth phase. Not the least of the challenges ahead are the underlying economic factors driving an entire industry.

This work reports the statistical preference for the generalized logistic model based on information theory and endeavors to describe the complex system underlying the evolution of the computer processor through the lens of density, which highlights transistor miniaturization. The standard measure of integrated circuit complexity may be unsuitable for understanding processor evolution because the number of transistors on a chip is coupled to the size of the chip. Moreover, the statistical properties of the modeling of time series data are generally neglected in the literature [92]. For example, the Moore's Law stepwise exponential model suffers from autocorrelation and underestimates 33% of the data because of the heteroscedasticity of the data. Unfortunately, data are unavailable from many other studies for independent testing for the presence of discontinuities embedded within the empirical data.

To conclude, we show that transistor density dynamics are characterized by a series of accelerations and decelerations that provide an alternative view of Moore's Law. The bi-logistic model statistically outperforms Moore's Law even though model complexity is compounded. Information-based statistical testing substantiates this model being more parsimonious than Moore's Law. Further, the multilogistic performs even better with an order-of-magnitude lower variance offsetting the risk of overfitting, notwithstanding the much higher complexity. Additionally, estimating parameter values for the characteristic times and the midpoints directly from the data minimizes the issue of overfitting. These findings cast doubt on the hypothesis of an exponential process in transistor evolution and indicate that more complex dynamics are at play.

Analysts have attempted to apply Moore's Law to other areas of technology, including DNA sequencing [93] and photovoltaics [94], among others [95]. Our revisiting of microprocessor evolution emphasizes the importance of choosing the right Y-axis. The evolution of computing may be as deeply understood through miniaturization and chip density as through chip count.

## Supporting information

**S1 Table. Data collected for the analysis is presented here.**
(DOCX)

## Acknowledgments

Portions of this work were prepared by S. Atler and B. Weizman and were submitted in partial sfulfillment of the requirements for B.Ed. certification at Tel Hai Academic College, Israel. We thank Prof. Tom O'Haver and David Laws and ComputerHistory.org for their helpful insights. JHA thanks JH Baker for questions about changing speed of technical evolution.

## Author Contributions

**Conceptualization:** David Burg, Jesse H. Ausubel.

**Data curation:** David Burg.

**Formal analysis:** David Burg.

**Investigation:** David Burg, Jesse H. Ausubel.

**Methodology:** David Burg.

**Supervision:** Jesse H. Ausubel.

**Writing – original draft:** David Burg, Jesse H. Ausubel.

**Writing – review & editing:** David Burg, Jesse H. Ausubel.

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
