## [Decision Letter · Decision Letter 0]

14 Jun 2021

PONE-D-21-15557

Moore’s Law Revisited through Chip Density

PLOS ONE

Dear Dr. Burg,

Thank you for submitting your manuscript to PLOS ONE. After careful consideration, we feel that it has merit but does not fully meet PLOS ONE’s publication criteria as it currently stands. Therefore, we invite you to submit a revised version of the manuscript that addresses the points raised during the review process.

We look forward to receiving your revised manuscript.

Kind regards,

Talib Al-Ameri, Ph.D

Academic Editor

PLOS ONE

Journal Requirements:

"Portions of this work were prepared by S. Atler and B. Weizman and were submitted in partial sfulfillment of the requirements for B.Ed. certification at Tel Hai Academic College, Israel. We thank Prof. Tom O’Haver and David Laws and ComputerHistory.org for their helpful insights. JHA thanks JH Baker for questions about changing speed of technical evolution. This work was supported by The Rockefeller University."

"This work was supported by The Rockefeller University."

"This work was supported by The Rockefeller University."

Reviewers' comments:

Reviewer's Responses to Questions

**Comments to the Author**

1. Is the manuscript technically sound, and do the data support the conclusions?

Reviewer #1: No

2. Has the statistical analysis been performed appropriately and rigorously? 

Reviewer #1: Yes

3. Have the authors made all data underlying the findings in their manuscript fully available?

Reviewer #1: Yes

4. Is the manuscript presented in an intelligible fashion and written in standard English?

Reviewer #1: No

5. Review Comments to the Author

Reviewer #1: The paper presents a very interesting mathematical analysis of transistor scaling from the invention of the transistor till transistor technology today. However, there are two major drawbacks of the paper/manuscript in its current form:

1) the recent analysed data do not include technology nodes from other semiconductor companies except Intel;

2) the most valuable relation between trends observed in the data and technological changes introduced into technology nodes in order to continue scaling or to continue advancement of the transistor technology is not present.

I would be happy to recommend a revised manuscript for publication if the two major drawbacks above are addressed. To clarify the two points:

1) the data of another semiconductor companies making digital semiconductor transistors from Samsung, TSMC, Global Foundries, IBM, Texas Instrument, SK Hynix, Micron Technology etc.

If the manufacturing of the semiconductor transistors turn to be to broad for the scope of the paper, the work should focus on the segment of the manufacturing of their choice and clearly state the focus.

2) a palette of so-called technology boosters was progressively introduced into the semiconductor manufacturing ranging from the change of Ge to Si, the change of metal interconnects, the change of various dielectrics, device architectures, the amendments of material properties like strain, etc.

In addition, I have the following points which needs to eb addressed too:

3) The (www.loglet.com, version 4). does not exists.

4) References are missing for these statements/claims:

i) Optimized parameter values were then obtained using a simulated annealing Monte Carlo–based genetic algorithm (www.loglet.com, version 4).

ii) Pearson correlations, tests for heteroscedasticity and autocorrelation (Breusch-Godfrey and Durbin-Watson tests, respectively) were performed on the linearized data (R, version 3.5.0).

5) The statements:

'R-squared was not included here'

'The corrected Akaike information criterion (AICc) allows model selection accounting for the complexity of the models'

have no sense.

Please, re-write and clarify.

6) The statement:

'The Automatic Maxima Detection software confirmed this result, showing peak growth rates coinciding with the inflection points

appears from nowhere/from a blue. Please, re-write and introduce.

7) Technical:

7a) To start a new section as 'Therefore, we defined transistor density...' is awkward. Please, re-write.

7b) There are a couple of formal errors as '(mean= 9 yrs).' A floating space. Please, carefully revised the whole text.

6. PLOS authors have the option to publish the peer review history of their article (what does this mean?). If published, this will include your full peer review and any attached files.

Reviewer #1: **Yes: **Karol Kalna

---

## [Author Response · Author response to Decision Letter 0]

28 Jul 2021

Reviewer #1 identifies two drawbacks

1. The recent analyzed data do not include technology nodes from other semiconductor companies except Intel. The reviewer suggests inclusion of the data of another semiconductor companies making digital semiconductor transistors from Samsung, TSMC, Global Foundries, IBM, Texas Instrument, SK Hynix, Micron Technology etc .

The reviewer adds: If the manufacturing of the semiconductor transistors turn to be to broad for the scope of the paper, the work should focus on the segment of the manufacturing of their choice and clearly state the focus .

We appreciate the suggestion but the suggestion would turn this paper, already quite long, into a monograph or book. We narrow the focus of the paper in the title, abstract and in the text by stating the inclusion of data from Intel/Fairchild and explaining our choice and challenges of a study of larger scope.

2.. …the most valuable relation between trends observed in the data and technological changes introduced into technology nodes in order to continue scaling or to continue advancement of the transistor technology is not present . …a palette of so-called technology boosters was progressively introduced into the semiconductor manufacturing ranging from the change of Ge to Si, the change of metal interconnects, the change of various dielectrics, device architectures, the amendments of material properties like strain, etc .

We agree with Reviewer #1 that this history is most interesting. We refer to several historical relationships in the Discussion. We are enthusiasts for such history and have added references to two excellent books, and an IEEE article as well as referencing online materials, which discuss exactly the kinds of developments to which Reviewer #1 refers. The purpose of our analysis is to understand the shape and mathematics of chip evolution; we do not claim to have done original research on particular causes. 

3. Citation to LogletLab software

We thank the reviewer for bringing this oversight to our attention. The modeling software is located at logletlab.com. This has been remedied.

4. References for 2 statements

The software citations were moved to the References section and cited in the text. We believe that common statistical procedures need no specific citations, and the original studies are, in any case, referred to in the software documentation.

5. Clarifications:

We clarified our decision to not include the R-square statistic and explained the use of AICc for model selection (as opposed to F-tests or comparison of goodness-of-fit measures which do not penalize the model for increased complexity and underestimating overfitting).

6. Clarify per Automatic Minimum Detection

This is not core to our results and was omitted.

7. Delete the word Therefore, start section “We defined…” ; check for any other spacing issues

We thank the reviewer and it has been corrected. We also made every effort to remove extraneous spacing.

---

## [Decision Letter · Decision Letter 1]

3 Aug 2021

Moore’s Law Revisited through Intel Chip Density

PONE-D-21-15557R1

Dear Dr. Burg,

We’re pleased to inform you that your manuscript has been judged scientifically suitable for publication and will be formally accepted for publication once it meets all outstanding technical requirements.

Kind regards,

Talib Al-Ameri, Ph.D

Academic Editor

PLOS ONE

Additional Editor Comments (optional):

It will be interesting if you compare the observation of Nvidia’s Jensen Huang with Moore’s Law, (optional).

https://spectrum.ieee.org/move-over-moores-law-make-way-for-huangs-law.

Reviewers' comments:

Reviewer's Responses to Questions

**Comments to the Author**

1. If the authors have adequately addressed your comments raised in a previous round of review and you feel that this manuscript is now acceptable for publication, you may indicate that here to bypass the “Comments to the Author” section, enter your conflict of interest statement in the “Confidential to Editor” section, and submit your "Accept" recommendation.

Reviewer #1: All comments have been addressed

2. Is the manuscript technically sound, and do the data support the conclusions?

Reviewer #1: Yes

3. Has the statistical analysis been performed appropriately and rigorously? 

Reviewer #1: Yes

4. Have the authors made all data underlying the findings in their manuscript fully available?

Reviewer #1: Yes

5. Is the manuscript presented in an intelligible fashion and written in standard English?

Reviewer #1: Yes

6. Review Comments to the Author

Reviewer #1: I am happy with the revised version of the manuscript.

Come on, machine, I have nothing else to write so do not be silly.

7. PLOS authors have the option to publish the peer review history of their article (what does this mean?). If published, this will include your full peer review and any attached files.

Reviewer #1: **Yes: **Karol Kalna

---

## [Editor Report · Acceptance letter]

9 Aug 2021

PONE-D-21-15557R1 

Moore’s Law Revisited through Intel Chip Density 

Dear Dr. Burg:

I'm pleased to inform you that your manuscript has been deemed suitable for publication in PLOS ONE. Congratulations! Your manuscript is now with our production department. 

Kind regards, 

on behalf of

Dr. Talib Al-Ameri 

Academic Editor

PLOS ONE